# Assessing Infection Risks among Clients and Staff Who Use Tattooing Services in Poland: An Observational Study

**DOI:** 10.3390/ijerph17186620

**Published:** 2020-09-11

**Authors:** Anita Gębska-Kuczerowska, Izabela Kucharska, Agnieszka Segiet-Swiecicka, Marcin Kuczerowski, Robert Gajda

**Affiliations:** 1Collegium Medicum, Cardinal Stefan Wyszyński University, Kazimierza Wóycickiego 1/3, 01-938 Warsaw, Poland; 2National Institute of Public Health, Chocimska 24, 00-791 Warsaw, Poland; 3Chief Sanitary Inspectorate, Targowa 65, 03-729 Warsaw, Poland; i.kucharska@gis.gov.pl; 4Faculty and Department of Experimental Physiology, Medical University of Warsaw, Żwirki i Wigury 61, 02-091 Warsaw, Poland; agnieszkasegiet@gmail.com; 5Hospital Praski, Aleja Solidarności 67, 03-401 Warsaw, Poland; mkuczerowski@vp.pl; 6Gajda-Med Medical Center, ul. Piotra Skargi 23/29, 06-100 Pułtusk, Poland; gajda@gajdamed.pl

**Keywords:** tattoo, infection risk, blood-borne infection, needlestick injury, Poland

## Abstract

Across cultures and generations, people have tattooed their bodies. Although blood-borne infections from tattooing have been reduced, certain service aspects remain improperly managed. We assessed the infection risks associated with tattooing by conducting a cross-sectional study (2013–2014) in Poland using an anonymous questionnaire survey. Scoring procedures for blood-borne infection risks for tattooists and their clients were used. Overall, 255 tattooists were interviewed. A quasi-random selection of tattoo parlors was based on a service register. Knowledge, attitudes, and behavior regarding blood-borne infection risks were assessed using a questionnaire. Simultaneously, tattoo centers were audited. Tattooing had a higher infection risk for tattooists than for clients. Approximately 50% of respondents underwent training on postexposure procedures, which constituted almost one in five of the reported needlestick/cut injuries sustained while working. Furthermore, 25.8% had no knowledge regarding risk from reliable sources, and 2.1% had not broadened their knowledge. Tattooists and their clients are at a risk of infection, and knowledge concerning infection risks remains an underestimated preventative factor. Service quality surveillance and creation of a register for tattoo-related complications may help assess the scale of this public health issue. However, a lack of these records implies the challenges in developing effective organizational and legal protections.

## 1. Introduction

In various cultures and across the ages, people have tattooed their bodies. Approximately 24% of citizens in the United States and approximately 12% of people in Europe have a permanent tattoo [1,2]. Superficial knowledge concerning the risks of tattooing can facilitate positive decision-making to acquire tattoos; therefore, to minimize this risk, only adults can undergo a tattooing procedure in some countries. Renzoni et al. [3] reported that despite the risks, adults opting for a tattoo cannot make a fully informed decision when giving informed consent in a manner similar to patients using medical services. Moreover, there is no current register in Italy to record complications due to tattooing, and there is a need for harmonized supervision in other EU countries. In one study, >50% of the people who were tattooed reported dissatisfaction with their tattoos [4]. Poorly considered decisions often result in attempts to remove a tattoo, AND adverse events and complications have been reported involving tattoo removal procedures [2]. Complication risks to clients are associated with the use of dyes, which have varying effects on the immune response. The risk of complications can be further increased owing to unintentional pathogenic infection [5,6,7]. Reducing the infection risk via strict adherence to asepsis guidelines should be maintained from the start of the tattooing process until wound healing. Providing appropriate information concerning the risks of infection and prevention should be mandatory to ensure that clients are fully informed before they consent [8]. From a public health perspective, Popalyar et al. [9] identified key areas for activities within the beauty service industry where preventative measures were necessary to avoid infections. These activities included education and training, a regulated infrastructure of services, and insurance of client safety, and a striving for improvement in the knowledge and skills of employees. In many countries, issues arising from the lack of a formal, permanent oversight system for beauty services remain unresolved (surveillance) [3,9]. 

This study aimed to assess the risk of infection transmission when performing a tattoo according to modifiable variables and a synthetic approach to risk. This study is the first to conduct a country-wide survey by using direct interviews with professionals while simultaneously auditing the service sites for tattooing.

## 2. Materials and Methods 

Initially, we proposed to conduct research in every province of Poland involving 30 tattoo parlors. However, owing to dynamic changes in the beauty service market, 255 (53%) of the 480 initially planned audits and interviews were conducted. We administered an anonymous questionnaire survey and audits in 16 provinces. All provinces were covered by the same branches of administration and legal authorities. A questionnaire was developed together with the State Sanitary Inspection (local governmental administration), and we consulted professional groups and specialists in infectious diseases and epidemiology while preparing the questionnaire. The selection of service points was quasi-random and based on a register available from the State Sanitary Inspection. Between November 2013 and March 2014, 255 people performing tattoos and permanent makeup were interviewed by inspectors and had their work sites audited. All discrepancies in questionnaires’ answers were explained simultaneously. The staff and work site questionnaires were divided into parts with open and closed questions. Participation in the study was voluntary and anonymous, and informed consent was obtained from all participants prior to commencing the interviews. The study proposal and design were accepted by the “Project HCV” Steering Committee and were undertaken at tattoo centers by the State Sanitary Inspection staff during their routine duties.

### Statistical Analysis

We analyzed and identified risk factors for blood-borne infections concerning staff performing beauty services and their clients. Only the respondents who performed tattoos and who completed all parts of the survey (i.e., responded to at least one question in each part of the survey) were considered in the analysis (i.e., 233 participants). Missing data were supplemented by assuming the worst answer in the context of risk assessment after consultation with staff who collected the data on site (those who gave the questionnaires). Data imputation was performed to use as many observations as possible because the computation of a risk score requires complete data. Data imputation was performed for 204 of 13,281 records (1.5% of records) and allowed the inclusion of 47 of 233 (20.2%) participants who would otherwise have been excluded because of missing data (all data with discrepancies were confirmed by inspectors as a worst-case scenario).

Owing to the lack of available data on blood-borne infections transmitted during tattooing, supervised statistical methods could not be used, and unsupervised methods based on an expert analysis was applied. Based on epidemiology experts’ opinions, the following scoring procedure for the risk of transmission of blood-borne infections was implemented. Particular survey questions were assigned a value to quantify the effect on the risk of infection transmission. This aspect of risk scoring was conducted externally by infection control experts (epidemiologists). Assigning values for risk owing to particular factors (questions) was arbitrary and was determined by a team of epidemiologists who undertook the risk assessment according to general knowledge on infection transmission and medical specifications as adapted for a nonmedical service sector. The procedure was performed separately regarding risk for staff and risk for clients. Questions in the questionnaire were grouped and referred to particular areas of risk modifiers: (1) applied disinfected/sterilized equipment, (2) infection control knowledge, (3) hand hygiene and the use of gloves, (4) use of sterile needles, and (5) needlestick injuries. Implemented scoring for risk was built as a linear model with coefficients for particular variables (questions) defined by weights assigned to the questions in the expert analysis. By using this risk scoring procedure, the risk scores for each respondent were calculated. The results were also converted into a 0%–100% scale across all ranges (risk for staff and risk for client) and areas of risk modifiers. The results (i.e., both calculated risk score and risk score converted to a 0%–100% scale) were analyzed descriptively. Comparison of the calculated risk scores between areas of risk modifiers enabled us to compare the effect of particular areas on the total risk of transmission of infections. A comparison of the risk scores converted into a 0%–100% scale allowed us to identify specific areas where the risk of transmission of blood-borne infections had relatively increased and areas that required further educational intervention. Within the identified areas of increased risk for the transmission of infection, we conducted a detailed data analysis, and descriptive statistics for raw survey data and for variables derived from raw data are presented in Table 1 and Table 2. 

Descriptive methods were used to present and analyze the results. In the descriptive statistics for continuous variables, the number of observations, as well as the mean, standard deviation, minimum, maximum, and median (along with 25% and 75% quantiles), is reported. For category variables, the number of observations and percentages for each category are reported.

## 3. Results 

Regarding education, 51.4% of the respondents had secondary education, 17.7% had vocational secondary education, 11.1% had higher education, 9.1% had postsecondary and bachelor’s education, and 1.6% had elementary education. The average time spent in formal education was 9.5 years (standard deviation (SD), 6.0; min–max, 0.5–33.0 years; median, 8.0; interquartile range (IQR), 5.0–14.0). The largest age group comprised individuals aged between 26 and 35 years (46.5%), whereas the average age estimated from grouped data was 34.9 (SD, 7.4; median, 34.0; IQR, 24.7–35.5) years. Among the 233 respondents included in the study, 100% of them performed tattoos only, and 75.0% were male.

## 4. Discussion

This study used a risk scoring procedure to assess the total risk for both tattooing staff and their clients to provide a synthetic view of the epidemiology of infections by identifying the areas of infection risk and by undertaking a detailed risk analysis. The novel contribution of this study is the proposed risk scoring methodology. However, it is also the main limitation of this study because the risk of infection has not been directly tested (measured) during tattooing procedures. During the first stage of survey development, consultations with a team of epidemiologists and tattooists were helpful. This team determined a reference value of risk based on their practical knowledge and reported studies. For a holistic approach to health risk assessment and given its multifactorial nature, many public health practitioners have developed synthetic rates of risk assessment. For example, this holistic approach is used for health status measurements and for determining the quality of services in the healthcare sector [10,11]. A global approach to risk in research refers to the evaluation of service systems in terms of organization, equipment, knowledge, and staff behavior and attitude. Evaluations include selected raw data obtained from questionnaires administered by service inspection companies across all of Poland. In our study, provincial level audits were conducted by the State Sanitary Inspection. Simultaneously, responses to questionnaires were collected from tattooists who indicated the most important risk areas, which was extended by the analysis of data on knowledge, applied equipment, disinfection and sterilization, and sterile needlestick injuries at sites.

We found that performing tattoos, paradoxically, entailed a higher risk for staff than for clients (Table 3). Staff are particularly aware of the necessity to protect clients and may sometimes neglect their own safety; therefore, they often suffer the consequences of occupational risk. The higher risk to staff was due to the frequency of occupational activities, risk of unintentional needlestick injury, and limited knowledge regarding postexposure prophylaxis. Beauty procedures may increase risk of infection when they are required to perform a process frequently. Similar to patients with chronic diseases within the medical sector, infection risks increase with the frequency of hospitalizations and frequency of performing invasive medical procedures [12,13,14]. One study investigating infection risk factors for patients aged ≥55 years found that 37% of new blood-borne infections with hepatitis B and hepatitis C were associated with injecting medication, and 8% were associated with hemodialysis [15]. One epidemiological study showed that the blood-borne infection risk was also high among drug-addicted individuals with high-risk behaviors, including intravenous drug use [16]. During tattooing, sterile disposable equipment and sterile dyes reduce infection risks; however, it is also important to be aware of other possible actions that may cause infection. Infections may be due to incorrect procedures or not maintaining aseptic techniques from the start of tattooing until wound healing [17,18]. In terms of epidemiological safety, the application of sterile and disposable equipment is essential. Disposable equipment does not guarantee sterility, and equipment taken from previously opened multipacks has been shown to be biologically contaminated [19]. One study found that in approximately 20% of the packaging used to contain tattoo ink that had been marked as sterile, contamination occurred because of microbiological pathogens. Package containing ink has been shown to be contaminated when it is stored for a long time after opening, or when the package had been stored in suboptimal conditions, which can cause the proliferation of pathogens or the spread of pathogens into the sterile content of the container [4,20]. Medically, tattooing involves other risks [12,21,22,23,24,25]. In a composition analysis of the dyes used in tattooing, some dyes contained mutagenic and carcinogenic compounds [26,27]. Some dyes cause local and distant reactions, and microorganisms may spread or remain localized at the site of the “injections” [4,28]. There have also been reports of sepsis and endocarditis with the transmission of infection as a consequence of tattooing [29,30,31].

In our study, we assessed the respondents’ knowledge of the risks of infection. The tattooists indicated (in order of importance) the following areas where infection risk could be reduced: applying disinfected/sterilized equipment, infection control knowledge, needlestick injuries, use of sterile needles, hand hygiene, and use of gloves during tattooing. According to the epidemiologists, staff transmission infection risk could be reduced in the following areas, which is presented in order of importance: infection control knowledge, disinfected/sterilized equipment, needlestick injuries, use of sterile needles, hand hygiene, and use of gloves during tattooing (Table 4 and Table 5). Tattooists’ rating of knowledge requirements did not indicate a significant understanding of this requirement. The importance of relevant knowledge development in many areas has been emphasized in many recommendations [32,33,34]. The primary importance of such knowledge does not reduce the importance of the other factors. In this study, tattooists correctly indicated the epidemiological chain elements of infection risks, but this information is based more from their clients’ perspective than from their own perspectives. Considering that 50% of the respondents participated in postexposure procedure training and that nearly one fifth of respondents reported needlestick injuries/lacerations while working, it appears that major educational and organizational requirements remain to be addressed. Despite obtaining information concerning clients’ infectious diseases, the tattooists remained at risk of transmission of infection (Table 1, Table 2 and Table 5). In tattoo parlors, owing to infection risks, it is recommended that a register of needlestick injuries be created, along with updated guidelines on postexposure procedures, injury prevention, and methods outlining sharps disposal procedures, such as those in medical and dental services [32]. However, there has been a limited application of guidelines for preventing infections (rules of asepsis) from the medical service sector to tattooing services; the beauty services market requires detailed analysis and recommendations to guarantee health safety [9].

The tattooists indicated that clients have the same areas of risk as themselves. According to the tattooists, disinfected/sterilized equipment was considered more important in the risk assessment of infections, whereas the epidemiologists in our study attributed a lesser value to this (Table 5). In relation to the tattooists, this finding can be attributed to the fact that not all tattoo sites used decontamination procedures or did not consider a need for the outsourcing of sterilization services (Table 6).

Owing to increasing requests for tattoos and the potential health risks due to early or late complications, most public health specialists have begun to focus on a need to raise awareness regarding infection risks, in addition to issues with regard to the monitoring and enforcement of aseptic techniques [9]. This topic requires ongoing attention from nonmedical professionals, to make certain that all procedures concerning the epidemiological chain of infections are clinically well known and implemented [34]. Matters regarding professional responsibility are considered in education programs; for example, physicians are held accountable for their treatment decisions and are required to update their knowledge [35,36,37]. However, among other professional groups whose work can have a substantial effect on client health, the relevant professional guidelines do not always require updating knowledge within the scope of health risk. In the current study, the respondents most often participated in training organized by companies offering aseptic products on disinfection and sterilization (61.4%) and less in training concerning prophylaxis of blood-borne infections (41.6%). Only 35.6% of the respondents were trained within the scope that guarantees full hazard safety. The introduction of infection control curricula on an e-learning platform has been reported to have increased accessibility to education without payment and without the marketing of products dedicated to tattooing services [33]. Before the provision of this professional e-learning platform, the main source of knowledge for tattooists was information obtained from the internet (85.8%) and from other tattooists (76%). Additionally, respondents broadened their knowledge from medical textbooks (57.5%), whereas 2.1% of respondents made no attempt at self-education. These results showed that despite insufficient knowledge concerning the risk of infection transmission, there was a high percentage of staff using disposable equipment (99.1%). The use of disposable equipment decreases infection risks but is insufficient in itself to guarantee safety. A high standard of sanitary services was reported in 12% of the tattooists who also applied disinfection and sterilization, in addition to using disposable equipment. Respondents used external sterilization services more often than disinfection. The benefits of managing equipment appropriately and of using external services (for example, sterilization of equipment by small medical sterilization units in terms of Health Technology Assessment analyses) have been presented by Dehnavieh et al. [38] Other studies using economic analyses in North America, Europe, and Asia have highlighted the potential market demands for the sterilization process in outsourcing services. The dynamic development of the market for sterilization outsourcing in relation to hospitals has also been reported and that consumers in many countries are mainly large entities [39]. Therefore, relevant guidelines need to consider this aspect of growing demand, as well as other organizational and legal factors. During an audit of tattoo parlors in Poland, the employees of the State Sanitary Inspection Department drew attention to the fact that effective sterilization does not guarantee the sterility of equipment and that other factors are also of key importance, for example, storage methods, expiry dates, and the maintenance of aseptic techniques when opening sterile packaging. The reuse of disposable equipment and re-sterilization has practical and safety limitations [40]. Another breach in the aseptic technique during tattooing (even when using disposable equipment) involves the frequent use of the same dyes on multiple clients, with increased likelihood of cross-contamination during tattooing [41,42].

The disinfection of equipment was undertaken more frequently by tattooists than sterilization; tattooists often used disinfection preparations, but their use does not necessarily make disinfection effective. State Sanitary Inspection employees indicated that it has not always been possible to verify the concentrations of disinfection fluids used in tattoo parlors or to determine how long they had been stored on site. Therefore, a good knowledge of decontamination methods is essential. Many countries’ laws and EU recommendations highlight the risk of tattooing. In particular, EU recommendations use science-based evidences to propose solutions for the problem within the EU (RES AP 2008 (1) and further amendments) [2]. Considering the results of this analysis, we suggest supplementing these recommendations with an active supervision of procedures, state of knowledge, and people practicing this profession.

## 5. Conclusions

Both tattooists and their clients have a risk of blood-borne infections. Professional tattoo/permanent makeup staff are also at risk of blood-borne infections because of the frequency of performing the service and an accumulation of risk factors. Considering the health risks in the beauty service market, information on tattooing and other procedures (including beauty procedures) is routinely obtained by physicians during collection of medical history from tattooed patients. In contrast, the risk of blood-borne infections is rarely perceived as an occupational risk of tattooists, which our risk scoring analyses indicated as a neglected aspect of risk assessment.

Approximately one in five (18.9%) tattooists in our study reported having needlestick injuries and lacerations while undergoing their work, and only one-third (35.6 %) had attended training concerning postexposure procedures, infection prophylaxis, and disinfection and sterilization methods and procedures. Knowledge regarding the risk of blood-borne infections remains an underestimated factor in relation to protecting both tattooists and clients. Among the respondents, 21.9% did not obtain their knowledge from reliable sources but from the internet, leaflets, or from other people, and 2.1% of the tattooists made no effort to extend their knowledge. Regular oversight for service quality and a register of complications arising from tattooing would provide an opportunity to assess the risk and scale of tattoo-related health issues from a public health perspective. From this perspective—given that there are no registers at present for both the adverse effects as well as accreditation procedures (e.g., staff/site updated certifications)—currently, it is not possible to readily apply organizational and legal regulations to address these issues. The project findings were used to tailor education programs for this occupational group. The training program and campaign significantly increased the risk awareness and knowledge of clients and tattoo artists, and our project has started a certified education program in the years 2012–2017 in Poland.

## Figures and Tables

**Table 1 ijerph-17-06620-t001:** Areas of relatively increased risk concerning detailed knowledge among participants.

Variable	Category	Total
Identify the infections that can be transmitted during tattooing	Viral (%)	216 (92.7)
	Bacterial (%)	221 (94.8)
	Fungal (%)	203 (87.1)
	Parasitic (%)	121 (51.9)
	All of the pathogens listed above (%)	118 (50.6)
Does the sterilization process totally destroy…?	Viruses (%)	226 (97.0)
	Bacteria (%)	226 (97.0)
	Fungi (%)	225 (96.6)
	Parasites (%)	206 (88.4)
	All of the pathogens listed above (%)	203 (87.1)
Does the disinfection process destroy...?	Viruses (%)	190 (81.5)
	Bacteria (%)	212 (91.0)
	Fungi (%)	206 (88.4)
	Parasites (%)	157 (67.4)
	All of the pathogens listed above (%)	136 (58.4)
Identify the methods that prevent the transmission of infection, e.g., viral infection.	Sterilization (%)	224 (96.1)
	Disinfection (%)	197 (84.5)
	Using disposable equipment (%)	231 (99.1)
	Hand hygiene (%)	227 (97.4)
	Washing surfaces (%)	223 (95.7)
	Washing tools (%)	194 (83.3)
	All of the procedures listed above (%)	173 (74.2)
Have you ever received training with regard to any of the topics listed?	Prophylaxis of blood-borne infections (%)	97 (41.6)
	The rules of disinfection and sterilization (%)	143 (61.4)
	Postexposure procedure (%)	117 (50.2)
	Yes, I have been trained in all the topics listed (%)	83 (35.6)
What have been your main sources of information concerning blood-borne infections, disinfection, sterilization, and postexposure procedures?	From other people performing tattoos (%)	177 (76.0)
	Internet (%)	200 (85.8)
	Medical textbooks (%)	134 (57.5)
	Leaflets and social campaigns (%)	117 (50.2)
	Courses for beauticians (%)	101 (43.3)
	Summary: courses for beauticians or medical textbooks (%)	173 (74.2)
	Other people, internet, and leaflets	51 (21.9)
	I have not broadened my knowledge (%)	5 (2.1)
Do you ask your clients about past or current viral infections? (YES response, %)		172 (73.8)

**Table 2 ijerph-17-06620-t002:** Needlestick injuries.

Variable	Category	Total
Have you ever pricked/injured yourself with a needle while at work? (Yes responses, %)		44 (18.9)

**Table 3 ijerph-17-06620-t003:** Tattoo risk scoring results * and converted scores **.

Variable	N	Mean	SD	Min	Max	Median	Q1	Q3	No Data
Risk to a person performing a procedure (%)	233	16.55	8.87	0.00	59.05	15.40	9.84	22.22	0
Risk to a client (%)	233	12.88	7.32	1.33	65.60	12.00	7.60	16.50	0
Risk to a person performing a procedure (points)	233	3.48	1.86	0.00	12.40	3.23	2.07	4.67	0
Risk to a client (points)	233	3.22	1.83	0.33	16.40	3.00	1.90	4.13	0

Note: * Calculated Score (risk scoring for staff, 0–21-point scale; risk scoring for clients, 0–25-point scale). ** Converted scores (scale 0%–100%).

**Table 4 ijerph-17-06620-t004:** Scoring the risk to a person performing a tattoo or risk modifiers.

Variable	N	Mean	SD	Min	Max	Median	Q1	Q3	No Data
Equipment; disinfection/sterilization (points)	233	0.90	0.37	0.00	3.00	1.00	1.00	1.00	0
Equipment; disinfection/sterilization (%)	233	29.90	12.29	0.00	100.00	33.33	33.33	33.33	0
Infection control knowledge (points)	233	2.12	1.52	0.00	6.86	2.00	0.79	3.13	0
Infection control knowledge (%)	233	19.31	13.86	0.00	62.34	18.18	7.21	28.48	0
Hand hygiene and use of gloves (points)	233	0.01	0.09	0.00	1.00	0.00	0.00	0.00	0
Hand hygiene and use of gloves (%)	233	0.86	9.24	0.00	100.00	0.00	0.00	0.00	0
Applied sterile needles (points)	233	0.07	0.30	0.00	2.00	0.00	0.00	0.00	0
Applied sterile needles (%)	233	1.72	7.52	0.00	50.00	0.00	0.00	0.00	0
Needlestick/cut injuries (points)	233	0.38	0.78	0.00	2.00	0.00	0.00	0.00	0
Needlestick/cut injuries (%)	233	18.88	39.22	0.00	100.00	0.00	0.00	0.00	0

**Table 5 ijerph-17-06620-t005:** Scoring of client risk according to particular areas or risk modifiers.

Variable	N	Mean	SD	Min	Max	Median	Q1	Q3	No Data
Equipment; disinfection/sterilization (points)	233	0.90	0.37	0.00	3.00	1.00	1.00	1.00	0
Equipment; disinfection/sterilization (%)	233	29.90	12.29	0.00	100.00	33.33	33.33	33.33	0
Infection control knowledge (points)	233	1.86	1.38	0.00	6.00	1.73	0.67	2.80	0
Infection control knowledge (%)	233	18.62	13.77	0.00	60.00	17.33	6.67	28.00	0
Hand hygiene and the use of gloves (points)	233	0.39	0.57	0.00	6.00	0.33	0.00	0.33	0
Hand hygiene and the use of gloves (%)	233	4.90	7.15	0.00	75.00	4.17	0.00	4.17	0
Applied sterile needles (points)	233	0.07	0.30	0.00	2.00	0.00	0.00	0.00	0
Applied sterile needles (%)	233	1.72	7.52	0.00	50.00	0.00	0.00	0.00	0

**Table 6 ijerph-17-06620-t006:** Equipment, disinfection, and sterilization—detailed results.

Variable	Category	Total
Application of disposable equipment, disinfection, and sterilization (%)	I use disposable equipment + disinfect + sterilize tools	28 (12.0)
	I apply one or two of the following methods: disposable equipment, disinfection, and sterilization	204 (87.6)
	I/we do not use disposable equipment + do not disinfect + do not sterilize	1 (0.4)
What type of sterilizing equipment do you use?	Autoclave (%)	52 (22.3)
	Sterilizer (%)	6 (2.6)
	Ultrasonic washer (%)	52 (22.3)
Who performs the disinfection of the tools used in your parlor?	I disinfect the tools (%)	140 (60.1)
	I have an agreement with an external entity (%)	9 (3.9)
	Disinfection is not performed (%)	80 (34.3)
Who sterilizes the tools applied in your parlor?	I sterilize the tools (%)	59 (25.3)
	I have an agreement with an external entity (%)	25 (10.7)
	Sterilization is not performed (%)	129 (55.4)
Do you use disinfecting fluids (Yes responses, %)?		230 (98.7)

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
