# Peer review of "Assessing Infection Risks among Clients and Staff Who Use Tattooing Services in Poland: An Observational Study"

_ijerph, 2020, doi:10.3390/ijerph17186620_

Round 1

Reviewer 1 Report

The article is very long and very complicated.

The proposed risk scoring methodology and method for statistical analysis is extremely complicated, more in particular the interpretation of the oucome and results; it is difficult to conclude about the reliability of the results, in particular the conclusions comparing risks for tattooists versus risks for clients.

The basic information (results of the questionaire) is quite interesting and gives a good view on the lack of knowledge of body art practitioners concerning the risks of the procedure and the need for better education.

The study goes back a few years (2013-2014); it would have been interesting to compare with the current status of education (2019). Has the knowledge improved in 2019? Have the hygienic conditions been changed?

There is no direct reference to the European resolution (RES AP 2008(1))which was available in 2013. There is no info concerning basic requirements of training (hygiene) and certification of tattooists in Poland.

The article could be reduced to the essential information, pointing out the major risks for tattooists and clients and give an indication how to improve the training.

Author Response

Reviewer 1:

Dear Reviewer:

We thank you for your thoughtful suggestions and insights. The manuscript has benefited from these insightful suggestions.

The manuscript has been rechecked and the necessary changes have been made in accordance with the suggestions. The responses to all comments have been prepared and given below.

 “Comments and Suggestions for Authors
-  The article is very long and very complicated. The proposed risk scoring methodology and method for statistical analysis is extremely complicated, more in particular the interpretation of the outcome and results; it is difficult to conclude about the reliability of the results, in particular the conclusions comparing risks for tattooists versus risks for clients.

Response:

1.We thank the Reviewer for this comment. While we agree that the proposed risk scoring methodology and method for statistical analysis are complex, we believe that they were necessary for comparing the risks for clients and staff. The scoring analysis method facilitates the “visualization of risk distribution” for both groups—clients and staff.

We have revised the manuscript to include clarification regarding the risk scoring methodology as follows:

“Considering the health risks in the beauty service market, information on tattooing and other procedures (including beauty procedures) is routinely obtained by physicians during the collection of medical history from tattooed patients. In contrast, the risk of blood-borne infections is rarely perceived as an occupational risk of tattooists, which our risk scoring analyses indicated as a neglected aspect of risk assessment.”

(page 9, lines 278-286)

  1. In accordance with the comments and suggestions, we have revised the Methods section to provide clarification to some of the issues raised. Specifically, we included information about the data collection, questionnaire, and how we dealt with missing or discrepant entries.

The revised portions of the manuscript are as follows:

“Between November 2013 and March 2014, 255 people performing tattoos and permanent makeup were interviewed by inspectors and had their worksites audited. All discrepancies in questionnaires’ answers were explained simultaneously. The staff and worksite questionnaires were divided into parts, with open and closed answers.”

(page 2, lines 71-75)

“Missing data were supplemented by assuming the worst answer in the context of risk assessment after consultation with staff who collected the data on-site (those who gave the questionnaires). Data imputation was performed to use as many observations as possible because the computation of a risk score requires complete data. Data imputation was performed for 204 of 13,281 records (1.5% of records) and allowed the inclusion of 47 of 233 (20.2%) participants who would otherwise have been excluded because of missing data (all data with discrepancies were confirmed by inspectors as a worst-case scenario).”

(page 2, lines 84-90)

The basic information (results of the questionnaire) is quite interesting and gives a good view on the lack of knowledge of body art practitioners concerning the risks of the procedure and the need for better education. The study goes back a few years (2013-2014); it would have been interesting to compare with the current status of education (2019). Has the knowledge improved in 2019? Have the hygienic conditions been changed?

Response:

We thank the Reviewer for this comment. We have included information about practical applications for the development of educational programs tailored for tattooists. This practical application was the goal of our project and it has been implemented with great success.

We have revised the manuscript as follows:

“Regular oversight for service quality and a register of complications arising from tattooing would provide an opportunity to assess the risk and scale of tattoo-related health issues from a public health perspective. From this perspective—given that there are no registers at present for both the adverse effects as well as accreditation procedures (e.g. staff/site updated certifications)—currently, it is not possible to readily apply organizational and legal regulations to address these issues. The project findings were used to tailor education programmes for this occupational group. The training programme and campaign significantly increased the risk awareness and knowledge of clients and tattoo artists, and our project has started a certified education programme in the years 2012–17 in Poland.”

(page 9, line 293- page 10, line 302)

There is no direct reference to the European resolution (RES AP 2008(1))which was available in 2013.

Response: We thank the Reviewer for this suggestion.

As requested, we have included the European resolution in the Discussion section as follows:

“Many countries' laws and EU recommendations highlight the risk of tattooing. In particular, EU recommendations use science-based evidences to propose solutions for the problem within the EU (RES AP 2008 (1) and further amendments) [2].”

(page 9, lines 270-274)

“There is no info concerning the basic requirements of training (hygiene) and certification of tattooists in Poland. The article could be reduced to the essential information, pointing out the major risks for tattooists and clients and give an indication how to improve the training.”

Response: We thank the Reviewer for this suggestion.

As requested, we have included as follows in conclusions:

“The project findings were used to tailor education programmes for this occupational group. The training programme and campaign significantly increased the risk awareness and knowledge of clients and tattoo artists, and our project has started a certified education programme in the years 2012–17 in Poland.”

(page  lines 299-302)

The information on basic requirements in trainings (hygiene knowledge) is included in tables.2-6.

Kind regards

Authors

Reviewer 2 Report

Excellent article with good basic research.  My only feeling is that the authors suggested that tattoo injuries should be registered, which is well-founded in their paper, but I would like the authors to also address if they feel that there should be certification and what that would entail.  In short, the paper is great, and I would like the authors to further expand their thoughts on solutions to their findings, even if it includes some conjectures.  

Author Response

Reviewer 2:

Dear Reviewer:

We thank you for your opinion. The manuscript has been rechecked and the necessary changes have been made in accordance with the suggestions.

The responses to comments have been prepared and given below.

Excellent article with good basic research. My only feeling is that the authors suggested that tattoo injuries should be registered, which is well-founded in their paper, but I would like the authors to also address if they feel that there should be certification and what that would entail. In short, the paper is great, and I would like the authors to further expand their thoughts on solutions to their findings, even if it includes some conjectures.

Response: We thank the Reviewer for the assessment of our manuscript and for this suggestion.

In accordance with the suggestion, we have expanded the Conclusion section as follows:

“Regular oversight for service quality and a register of complications arising from tattooing would provide an opportunity to assess the risk and scale of tattoo-related health issues from a public health perspective. From this perspective—given that there are no registers at present for both the adverse effects as well as accreditation procedures (e.g. staff/site updated certifications)—currently, it is not possible to readily apply organisational and legal regulations to address these issues. The project findings were used to tailor education programmes for this occupational group. The training programme and campaign significantly increased the risk awareness and knowledge of clients and tattoo artists, and our project has started a certified education programme in the years 2012–17 in Poland.”

(page 9, line 293-page 10, line 302)

Kind regards

Authors

Reviewer 3 Report

Dear Authors,

the content of the article ist very interesting. However, I had problems to follow the article. 

Mainly, the description of the method must be improved. In line 59 is written "by using direct interviews with professionals while simultaneously auditing the service sites for tattooing." But the description of the survey is not clear.

How do you collect the data? During an interview and the researcher fill out the questionnaire? In auditing and the researcher fill out the questionnaire? Or do the tattoist fill it out themselves?

It is not unusual to supplement missing data. It is not clear why you decided to choose the worst answer and not e.g. to use the mean. There must be an argument for this and which effect this means for the data.

I have some difficults to understand some information. For example: The 16 provinces have the same law for run a studio or is it from province to province difficult. Why are the number auf male interesting in the chapter Material and Methods, this is a result?

Please check the tables for correct data: for example see Table 5 last question.

Best regards

Author Response

xcv

Reviewer 3:

Dear Reviewer:

We thank you for your thoughtful suggestions and insights. The manuscript has benefited from these insightful suggestions.

The manuscript has been rechecked and the necessary changes have been made in accordance with the suggestions. The responses to all comments have been prepared and given below.

  1.  “ the content of the article ist very interesting. However, I had problems to follow the article. Mainly, the description of the method must be improved. In line 59 is written "by using direct interviews with professionals while simultaneously auditing the service sites for tattooing." But the description of the survey is not clear. How do you collect the data? During an interview and the researcher fill out the questionnaire? In auditing and the researcher fill out the questionnaire? Or do the tattoist fill it out themselves? It is not unusual to supplement missing data. ‘’

Response: We thank the Reviewer for this comment. In accordance with the comments and suggestions, we have revised the Methods section to provide clarification to some of the issues raised. Specifically, we included information about the data collection, questionnaire, and how we dealt with missing or discrepant entries.

The revised portions of the manuscript are as follows:

“Between November 2013 and March 2014, 255 people performing tattoos and permanent makeup were interviewed by inspectors and had their worksites audited. All discrepancies in questionnaires’ answers were explained simultaneously. The staff and worksite questionnaires were divided into parts, with open and closed answers.”

(page 2, lines 71-75)

It is not clear why you decided to choose the worst answer and not e.g. to use the mean. There must be an argument for this and which effect this means for the data.

            Response: We thank the Reviewer for this comment.

The revisions are the following:

“Missing data were supplemented by assuming the worst answer in the context of risk assessment after consultation with staff who collected the data on-site (those who gave the questionnaires). Data imputation was performed to use as many observations as possible because the computation of a risk score requires complete data. Data imputation was performed for 204 of 13,281 records (1.5% of records) and allowed the inclusion of 47 of 233 (20.2%) participants who would otherwise have been excluded because of missing data (all data with discrepancies were confirmed by inspectors as a worst-case scenario).”

(page 2, lines 84-90)

I have some difficults to understand some information. For example: The 16 provinces have the same law for run a studio or is it from province to province difficult.

Response: We thank the Reviewer for this comment.

All provinces have the same law regulations. As requested, we have revised the manuscript to clarify this issue and moved the demographic information to the appropriate section (Results section).

The revisions are the following:

“All provinces were covered by the same branches of administration and legal authorities.”

(page 2, lines 66-67)

Why are the number auf male interesting in the chapter Material and Methods, this is a result?

Response: We thank the Reviewer for this comment.

We removed this sentence (information) to Result sec.

“Among the 233 respondents included in the study, 100% of them performed tattoos only, and 75.0% were male.”

(page 3, lines 127-128)

Please check the tables for correct data: for example see Table 5 the last question

Response: We thank the Reviewer for this comment.

We have corrected the data mistake - presented percentage in Table 5.

“Do you use disinfecting fluids (Yes responses, %)?

230 (98.7)”

We thank the Reviewer for all suggestions. In accordance with the comments and suggestions, we have revised the information in sections: Method, Discussion, and Conclusion - by providing additional information and clarification.

                                                           Kind regards

Authors

Round 2

Reviewer 1 Report

The responses above are satisfactory for most of the remarks, "ACCEPT" but with recommendation of statisticians' opinion.

It would have been interesting to report the outcome of the education campaign >2017 and make a comparison with the results 2013-2014. or this could be subject for a second article.

It seems that even if you indicated you did, you forgot to include in the text a reference to the European resolution (RES AP 2008(1)?